# Immunothrombosis: Molecular Aspects and New Therapeutic Perspectives

**DOI:** 10.3390/jcm12041399

**Published:** 2023-02-09

**Authors:** María Marcos-Jubilar, Ramón Lecumberri, José A. Páramo

**Affiliations:** 1Hematology and Hemotherapy Service, Clínica Universidad de Navarra, 31008 Pamplona, Spain; 2CIBER-CV, Instituto de Salud Carlos III, 28029 Madrid, Spain; 3Instituto de Investigación Sanitaria de Navarra (IdiSNA), 31008 Pamplona, Spain

**Keywords:** immunothrombosis, thromboinflammation, tissue factor

## Abstract

Thromboinflammation or immunothrombosis is a concept that explains the existing link between coagulation and inflammatory response present in many situations, such as sepsis, venous thromboembolism, or COVID-19 associated coagulopathy. The purpose of this review is to provide an overview of the current data regarding the mechanisms involved in immunothrombosis in order to understand the new therapeutic strategies focused in reducing thrombotic risk by controlling the inflammation.

## 1. Introduction

The activation of the coagulation cascade represents a natural mechanism of defense, critical to keep physiological hemostasis in response to an infection or tissue damage in order to avoid blood loss. Nowadays, coagulation is considered a key player in the immune response, as it could prevent both viral and bacterial infections working in a bidirectional way with innate immune system [1]. However, in pathological situations, excessive activation of the coagulation cascade may occur, leading to a disseminated intravascular coagulopathy (DIC) and other thrombotic processes [2,3]. The link that connects both abnormal coagulation activation and innate immunity was first named immunothrombosis by Engelmann and Massberg in 2013 [4].

Classically, the coagulation cascade was described to be composed of two pathways, intrinsic and extrinsic, leading to a common final pathway, that originates the fibrin blood clot which is able to stop the bleeding that occurs after a vascular lesion. The intrinsic pathway begins with factor XII activation due to the interaction of blood with artificial or pathological surfaces that are negatively charged, such as DNA, RNA, polyphosphates, or atherosclerotic plaque components. The extrinsic pathway begins with tissue factor (TF) that makes a complex with factor VII at subendothelial level or over circulating immune cell membranes, as monocytes. However, in pathological situations, the pathogen recognition by immune cells amplifies procoagulant TF activity around 100 times [5]. Both acute and chronic inflammatory diseases promote dysregulation of hemostasis leading to aberrant clot formation, which is now termed immunothrombosis. Therefore, inflammation and coagulation are closely connected, what explains that immunothrombosis or thromboinflammation will characterize many clinical situations, such as sepsis or DIC, but also myocardial infarction, stroke, venous thromboembolism, or COVID-19-associated coagulopathy [3,4,6]. Molecular mechanisms involved in this process lead to TF activation and release and inflammasome activation [7].

## 2. Tissue Factor Initiates Immunothrombosis

TF is a 47 kd membrane receptor glycoprotein that plays an important role in coagulation activation in response to infectious agents or tissue injury. It is expressed in blood vessel adventitia and in circulating cells, such as monocytes and neutrophils [5]. In the physiological hemostasis, the exposed subendothelium releases TF to the blood flow. TF then interacts with factor VII and activated factor VII (FVIIa) forming a complex that activates the extrinsic pathway of coagulation and finally it is degraded by its natural inhibitor: tissue factor pathway inhibitor (TFPI). In infectious scenarios, pathogen associated molecular patterns (PAMPS), such as bacterial lipopolysaccharides, are detected by pattern recognition receptors (PRRs), such as TLR4, leading to TF induction at the mRNA level. This process takes place through NF-Kβ transcription factor activation in monocytes, neutrophils, endothelial, and epithelial cells [5,8]. Along this process, TF is decrypted due to lipidic membrane conformational changes, enhancing its procoagulant activity. Decryption process is needed to activate the TF cofactor in order to allow the binding of TF to FVIIa and FX. In vitro models show a faster union of decrypted TF to FVII/FVIIa compared to cryptic TF, but also demonstrate that decrypted TF can cleave both a peptidyl substrate and FX while cryptic TF can only cleave peptidyl substrate [9]. Additionally, there are two essential steps in the decryption process: the formation of a disulfide bond between unpaired cysteine residues 186 and 209 in cryptic TF and the exposure of phosphatidylserine on the cell surface of endothelial cells, neutrophils, and monocytes [9].

Then, immune cells release decrypted TF in a process called pyroptosis that provokes leaks in the cellular membrane and activates the inflammasome through the caspase pathways [7,10]. Following its release as extracellular vesicles, TF forms a high affinity complex with activated factor VII that leads to the activation of factor X to Xa, IX to IXa, and finally generating thrombin and the conversion of fibrinogen to fibrin. Additionally, thrombin activates platelets via protease activated receptors (PARs) that have a critical role in inflammatory processes through cytokine release [11,12] (Figure 1). Fibrin clot formation is enhanced by the presence of activated platelets and neutrophils that release extracellular traps (NETs) in a process called NETosis [13,14]. NETs spread thrombosis by retaining TF and extracellular vesicles, inducing endothelial dysfunction, and activating the intrinsic coagulation pathway due to its negatively charged surface [15]. In summary, PAMPs recognition by PRRs leads to TF release and decryption that increased its activity beginning the immunothrombosis/thromboinflammation process.

Additionally, platelet adhesion to injury site turns platelets into an active state that may act as a beginner for the TF alternative pathway [16].

In summary, published data show that TF, together with monocytes or neutrophils, is a crucial player during immunothrombosis as it can activate platelets [16].

## 3. Inflammasome Activation

Inflammasomes are molecular platforms conformed by several proteins that regulate the inflammatory response as antimicrobial mechanism defense. The presence of PAMPs and other dangerous molecular signals (DAMPs) activate caspases and induces pyroptosis and cellular death through gasdermins, which are the executors of inflammation and cell death. The so called non-canonical inflammasome activation is produced as a consequence of caspase-11 induction by LPS-associated gram-negative bacteria in different immune and nonimmune cells, including neutrophils, macrophages, and endothelial cells [17]. The caspase-11 activation promotes gasdermin D proteolysis, which provokes cell lysis named pyroptosis, which is a critical antibacterial defense mechanism that avoids pathogen dissemination [7,18,19].

Caspase-1 leads to the canonical activation of the inflammasome and provokes NLRP3 activation and IL-1β and IL-18 cytokine release. Caspase-1 is activated in the presence of several pathogens and at the same time activates gasdermin D, this activation promotes pyroptosis that originates in gaps in the cellular membrane, increasing TF-related procoagulant activity [8] (Figure 1).

Recent experimental and clinical studies have demonstrated that the canonical and non-canonical activation of the inflammasome are crucial for TF release by immune cells [17]. In this sense, several reports show that caspase-1 is present in human macrophages during sepsis. Moreover, Wu et al. [7] showed that canonical (by E. coli proteins) and non-canonical (by LPS) macrophage activation of the inflammasome induce TF release through pyroptosis, enhancing thrombosis and mortality in experimental models, while caspase-1 and 11 deletions block TF release [7]. Another recent study showed that caspase-11 and gasdermin D activation are essential for LPS related thrombosis, demonstrating that gasdermin D increases procoagulant TF activity through phosphatidylserine externalization, a membrane phospholipid that favors TF decryption and also catalyzed the activation of coagulation proteins, which may also contribute to an increased thrombotic profile [20]. Finally, in a mouse experimental model, caspase-1 and gasdermin D deletions protected the animals from thrombosis [21].

In conclusion, clinical and experimental evidence establish cellular death mediated by inflammasomes as a key trigger for immunothrombosis and microvascular thrombosis [22]. Two key signals are required for the coagulation activation mediated by the inflammasome: induction of TF protein and inflammatory caspase activation that induce TF release through pyroptosis [23] (Figure 1).

Recently, new mechanisms involved in the immunothrombosis process have been proposed. One of them is related to the STING protein (stimulator of the interferon response cGAMP interactor 1), expressed in fibroblasts and endothelial cells which responds to PAMPs and DAMPs. The STING protein has been previously implicated in the inflammatory response in sepsis-associated coagulopathy in experimental models. STING favors the procoagulant response through the regulation of the calcium release and gasdermin D proteolysis, facilitating TF release [24]. Another mechanism involves HMGB1 (high-mobility group box protein 1), a DAMPs that is increased in patients with sepsis or trauma and also in animals treated with LPS. In vitro, HMGB1 stimulates TF expression and release, contributing to sepsis induced by gram-negative bacteria after LPS linkage. HMGB1 transports LPS to the cytosol where it induces pyroptosis through the non-canonical inflammation activation mediated by caspase-11 [25].

## 4. Role of NETs in Immunothrombosis

NETs are extracellular structures composed of granule, nuclear, and mitochondrial constituents assembled on a scaffold of decondensed chromatin with antimicrobial proteins and peptides [26,27]. NETs act as a scaffold for platelets, red blood cells, extracellular vesicles, VWF, and TF [28]. NETs play a crucial role in host defense and pathogen clearance during infection [29]; however, dysregulation of NETs can lead to autoimmune, inflammatory, and pathological thrombotic disorders [26,27,28]. Histone proteins, that are one of the constituents of NETs (in addition to DNA), are potent DAMP molecules also capable of initiating a positive inflammatory feedback loop [26].

Nucleic acids activate coagulation, with RNA binding both factors XII and XI in the intrinsic pathway. RNA is present in fibrin-rich arterial thrombi. Furthermore, histones increase thrombin generation in a platelet-dependent manner. Histones activate platelets, and platelet activation, in turn, promotes coagulation, leading to platelet-rich microthrombi formation. Within thrombi formed in vivo, NETs colocalize with VWF [30]. Finally, NET fibers contain various other factors that can render them procoagulant, such as serine proteases that inhibit TFPI, and in addition, TF has been shown to be deposited on NETs [26,28].

It is interesting to note that although platelet and neutrophil interplay is crucial, depending on the original stimuli platelets, they are not always needed [31]. In fact, gram-negative bacteria are recognized by platelet LPS-TLR4 binding, inducing NET formation due to activated platelet interactions with neutrophils through surface proteins (CD62P, GPIbα and integrin αIIbβ3) [31,32,33,34]. However, LPS stimulation can also directly form NETs in the absence of platelets; as it occurs in some viral infections, such as COVID-19, that does not need platelet interplay to activate NETs, since neutrophils can recognize virus RNA through the TLR receptor, including the intracellular receptors, such as TLR7, TLR8, and TLR9. Following virus recognition, specific TLRs activate PAD 4 directly [35]. When this process occurs without control, thrombus formation appears and, if chronified, it can lead to tissue damage resulting in fibrosis, organ dysfunction, and death [26].

## 5. Role of the Von Willebrand Factor in Immunothrombosis

The von Willebrand Factor (VWF) is a glycoprotein release by endothelial cells or platelets with an important role in hemostasis [36]. Furthermore, leukocyte recruitment is mediated by VWF [37]. It is usually secreted in the form of ultra-large VWF (UL-VWF) multimers that are cleaved by ADAMTS13, a metalloproteinase, regulating its size and activity [38].

It is described that VWF is colocalized with NETs in venous and arterial thrombosis [39,40], inducing a proinflammatory and prothrombotic state. Some studies showed that NETs increased cytokines levels, such as IL-6 and reduce ADAMTs13 activity [30,41], leading to an increase in UL-VWF. In fact, NET-VWF binding can recruit platelets and leukocytes, thus playing a role in thrombotic microangiopathies, ischemic stroke, and COVID-19 disease [30].

## 6. Infectious Disease-Induced Immunothrombosis

Several studies have addressed the close interaction between sepsis and coagulopathy. Sepsis is characterized by a dysregulated immune system with an excessive production of cytokines, in response to infectious stimuli [42]. This dysregulation can lead to endothelial damage and upregulation of the TF expression, not only in endothelial cells, but also in other circulating cells [43]. During bacterial infections, non-canonical inflammasome activation is crucial in the cell surface TF release [44]. Additionally, pathogen and cytokines activate the endothelium, producing a prothombotic state promoting multimeric chains of UL-VWF release to endothelial cell surface where they will bind platelets [45]. Usually, UL-VWF is broken by ADAMTs13 into small multimers, however, in patients with sepsis, ADAMTs13 is reduced, being unable to break these long structures which originate in a string surrounded by platelets and neutrophils that are the bases for microthrombi formation [46]. Finally, during sepsis, several anticoagulant proteins are reduced, such as protein C, protein S, and antithrombin, favoring thrombin generation and fibrin formation [47].

There is growing interest in the immunothrombosis role in SAR-COV-2 (COVID-19) infection complications. Previously, in vitro studies demonstrated TF induction and decryption in response to different viruses, such as herpes simplex, Ebola, and HIV associated with different coagulopathy degrees and thrombosis [48,49]. However, there is a gap in the knowledge about the exact mechanism of TF induction and release upon viral infections [17].

It is known that COVID-19 induces an important coagulopathy in which multiple components are involved in immunothrombosis and play an essential role [50,51]. First, infected endothelial cells release VWF, attracting local neutrophils and platelets that promote inflammation and coagulation [17]. Moreover, COVID-19 provokes endothelial cell damage and apoptosis, leading to a decrease in antithrombotic activity [52]. In addition, both activated neutrophils and platelets are critical in NETs release which capture TF and TF positive micro-vesicles, triggering coagulation cascade activation [17,53]. A severe COVID-19 infection induces a hyperactivation of the immune system that leads to an uncontrolled release of cytokines, so-called cytokine storm. Among these cytokines, IL-6, interferon-γ (INF-γ), and IL-2 increase platelet production and TF expression on endothelial cells and monocytes. Moreover, IL-2 increased the procoagulant state by decreasing fibrinolysis [54]. Finally, COVID-19 induces canonical NLRP3 and non-canonical caspase-11 activation of the inflammasome [17,50,51]. In the end, a COVID-19 infection should be considered as an abnormal/dysregulated immunothrombosis syndrome involving pulmonary microcirculation [17,52].

## 7. Thromboinflammatory Conditions

### 7.1. Arterial Diseases and Immunothrombosis

The activating interplay of thrombosis and inflammation (thromboinflammation) has also been established as a major underlying pathway driving cardiovascular diseases, such as myocardial infarction and stroke. Innate immunity and platelet crosstalk promote plaque formation and rupture, also depending on the interaction of UL-VWF with NETs [54]. Decorated with histones and cytoplasmic and granular proteins, NETs exert cytotoxic, immunogenic, and prothrombotic effects accelerating disease progression. The inflammasome, through generation of active caspase-1, causes the release of the important vascular effector cytokine IL-1β. IL-1β, in turn supports the endothelial expression of adhesion molecules, such as E-selectin, favoring leukocyte recruitment. Activated platelets also stimulate TF synthesis and its release in vesicles from monocytes leading to thrombin generation and fibrin formation [54]. Additionally, NETs activate a complement system leading to increased endothelial damage and upregulate platelet activation [2].

Finally, clonal hematopoiesis (CHIP) is related to inflammation and cardiovascular diseases. CHIP mutations provoke a pro-inflammatory ambient that produce an increase in the thrombotic risk and favor plaque formation. These mutations produce functional alterations in neutrophils showing a prothrombotic profile. Among them, JAK2 mutations increase thrombotic risk through PAD4-dependent NETosis [55].

### 7.2. Autoimmune Diseases

Autoimmune disorders, such as inflammatory bowel disease (IBD) and systemic lupus erythematous (SLE), are associated with the increased risk of venous thromboembolism [43].

IBD presents an increase in proinflammatory immune cells (lymphocytes, neutrophils, and monocytes) and cytokines. During flares, IBD patients increase procoagulant factors and reduce antifibrinolytic proteins [56,57]; moreover, procoagulant TF-MPs are shed by platelets and leukocytes that may stimulate NETosis [58]. In addition, the protein C pathway is an important regulator of mucosal homeostasis maintaining endothelial cell function regulating inflammatory responses, leading to spontaneous gut inflammation when the protein C pathway is altered [59].

Patients with SLE characteristically present antiphospholipid antibodies that are related with the increased thrombotic risk in these patients [60]. However, they also present endothelial dysfunction, fibrinogen and cell adhesion molecules increase [61]. Furthermore, SLE patients have a reduced NETs clearance with an altered immune profile [62]. Finally, interaction between antiphospholipid antibodies and monocyte receptors facilitates TF decryption and promote coagulation [63].

## 8. Immunothrombosis as a New Therapeutical Target

Standard antithrombotic therapy with anticoagulants is highly effective, but it is associated with increased hemorrhagic complications because fibrin is an essential component of the physiological hemostasis [64]. Usually, they exert their effect, decreasing coagulation factor activities [17]. Several studies have shown that hemostasis can be spared with effective thromboprotection, among them it has been demonstrated that a modified heparin without anticoagulant activity blocks HMGB1, inhibiting the caspase-11 mediated pyroptosis, preventing sepsis, thrombosis, and mortality in mice, without increasing hemorrhagic risk [65]. It is possible that coagulopathy may be better prevented by inflammasome inhibition of the TF induction rather than by the coagulation factor inhibition [17,23].

New approaches are needed to improve unbalanced hemostasis without increasing the bleeding risk, we summarize in Table 1 the different therapeutic approaches for immunothrombosis:

(a)Targeting coagulation

Some treatments usually used as an anticoagulant and antiplatelet may exhibit different effects modulating immunothrombosis. In fact, heparin, one of the most common anticoagulants, at high concentrations, is able to degrade NETs and also neutralize histones in blood. Furthermore, antiplatelet drugs, such as aspirin or ticagrelor, inhibit NET release [26].

(b)Targeting NETs

Several targeted therapies, focused on different steps of NETosis, have been explored in recent years. DNase is a nuclease that breaks down the NETs DNA backbone; however, to exert its effects, it needs the addition of fibrinolytic agents. To date, DNase 1 has been approved as a nebulized treatment in cystic fibrosis and there are several trials assessing the efficacy and safety of intravenous DNases administration after thrombectomy [26].

Another target to disrupt NET formation and prevent its release is the enzyme PAD4 which is essential in the early stage of NET formation [66]. Recently, PAD inhibition has been gaining interest and several inhibitors have been developed; initially pan-PAD inhibitors were created, such as compounds derived from benzol-arginine; however, even though there are promising results in preclinical trials, some unwanted side-effects in human have appeared, preventing its used in clinical practice. Lately, directed PAD inhibitors, such as PAD4 inhibitor (GSK199 and GSK484), seem to offer better results, inhibiting PAD4 with high specificity [67].

(c)Targeting inflammation

The question that should be raised is: may the inflammasome and pyroptosis inhibition be a new strategy for immunothrombosis treatment? In this regard, several JAK pathway inhibitors, such as baricitinib, ruxolitinib, or tofacitinib, may reduce immunothrombosis [17]. Likewise, several STING inhibitors have been recently identified, such as nitrofurans, indole ureas, and acrylamides that, at least ex vivo, block TF induction in SARS-CoV-2 infected endothelial cells [68]. Moreover, NLRP3 inhibition with MCC950 dimmed the platelet activation in a rat sepsis model [69]. Dimetil fumarase is being investigated as an anti-inflammatory strategy for COVID-19 patients because it inhibits gasdermin D, exerting immunomodulatory effects, and has been shown to reduce pyroptosis in an experimental murine colitis model [70,71]. Several drugs focused on blocking inflammation through HMGB1 have been explored in preclinical settings, showing favorable results in controlling both sterile and infectious inflammatory conditions and DIC [72].

Finally, some anti-inflammatory drugs used in daily clinical practice can also reduce immunothrombosis. Colchicine, frequently used in gout treatment, reduced cardiovascular events in the COLCOT study, when administered to patients who had suffered a myocardial infarction [73]; this drug inhibits immunothrombosis by reducing NETosis and attenuates the NLRP3 activation [74].

(d)Targeting Complement

The complement system, part of the innate immune response against bacterial and viral infections, can be activated through classical pathways (antibody-antigen complex), alternative pathways (specific surface antigen), and lectin pathways (mannose residues on pathogen surface). All of these converge in a common pathway, including C3a and C5a production and C3b-initiated pathogen opsonization, ending in C5b-9 membrane attack complex formation, which results in target cell lysis [75]. In vitro, blocking C5a with a murine antibody decreased the cytokine response and viral replication in MERS-CoV infections. In humans, complement inhibition with eculizumab, an antibody against C5 that prevents the breakdown in C5a and C5b, have been used as treatment for thrombotic microangiopathy (TMA), which is a manifestation of different clinical scenarios characterized by abnormal complement activation. Recently during the COVID-19 pandemic, it has arisen as a new treatment opportunity as it reduces the innate immune-mediated consequences of a severe coronavirus infection [2,75].

**Table 1 jcm-12-01399-t001:** Therapeutic strategies for immunothrombosis.

Target	Drugs
Coagulation	
Anticoagulant	Low molecular weight heparin [66], fondaparinux [76].
Antiplatelets	Aspirin [77], ticagrelor [78].
NETS	Colchicine [73,74], heparin [79], aspirin [77], ticagrelor [78], DNASes [26].
Inflammation	
JAK-STAT pathway inhibitors	Baricitinib, ruxolitinib, tafacitinib [80].
STING inhibitors	Nitrofurans, acrylamides, indole ureas [68].
Inflammasome inhibitors (NLRP3)	MCC950 [71], colchicine [73,74].
Gasdermin D inhibitors	Dimetil fumarate [70,71].
HMGB1 inhibitors	Peptide p5779, m2G7, metformin, thrombomodulin [72].
Complement	Eculizumab [75].

## 9. Conclusions

In the last decade, there has been a huge advance in the molecular knowledge related to immunothrombosis development. Immunothrombosis is defined as a bidirectional interaction process between the innate immune pathway and coagulation. New strategies focused on reducing the thrombotic risk without increasing the hemorrhagic risk through inflammation control have been investigated. Among them, TF expression, inflammasome and NETs blockage are new attractive strategies to block the immunothrombotic process without the bleeding effects related to the traditional anticoagulants.

## Figures and Tables

**Figure 1 jcm-12-01399-f001:**
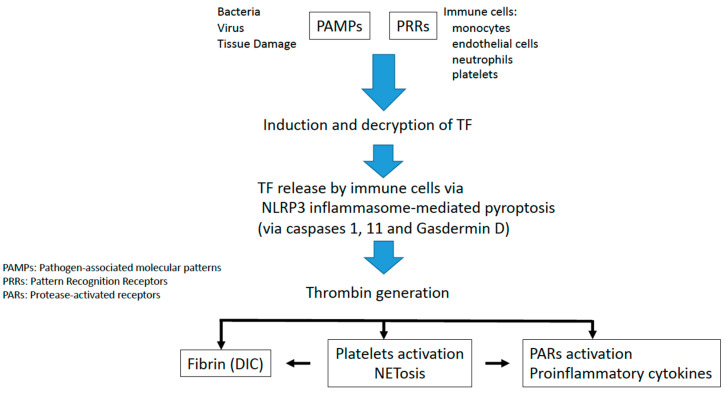
Molecular mechanism implied in immunothrombosis/thromboinflammation. Following the immune cell’s activations, TF decryption is induced leading to the TF release from these immune cells and also pyroptosis through the NLRP3 inflammasome. This process induces thrombin generation that provokes platelet activation and the NETosis process increasing fibrin formation and proinflammatory cytokine release.

## Data Availability

Not applicable.

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
