# Peer review of "Immunothrombosis: Molecular Aspects and New Therapeutic Perspectives"

_jcm, 2023, doi:10.3390/jcm12041399_

Round 1

Reviewer 1 Report

The manuscript is a well written short review which gives future perspectives. The paragraph referring to “new therapeutical target” could be more extensive. More details from the experimental studies should be described. 

Author Response

Thank you for your feed back, as required, we have extend the discussion about new therapeutical target and details about experimental studies, these can be seen in the new manuscript version that has been upload.

Reviewer 2 Report

1) Please describe more details in the pathophysiology of the viral-associated coagulopathy, especially in COVID-19-associated, and evidence of potential targeted therapeutic agents in previous studies.

2) The figure which explains more detail in pathophysiology of immunothrombosis, including particular mechanism of action of each potential targeted therapeutic agent should be added

Author Response

Thank you for your kind suggestions, we ask separately to each one:

  • Please describe more details in the pathophysiology of the viral-associated coagulopathy, especially in COVID-19-associated, and evidence of potential targeted therapeutic agents in previous studies.

As suggested by the reviewer pathophysiology of COVID 19 coagulopathy has been described more thoroughly in page 2.

  • The figure which explains more detail in pathophysiology of immunothrombosis, including particular mechanism of action of each potential targeted therapeutic agent should be added

We have tried to better explain the therapeutic target in a new table 1, that summary the target, the molecule and includes the references articles. But if even the improvement of this table, you still believe that a new image or a change in figure 1 including the therapeutical approach is needed we can create it.

Reviewer 3 Report

I read the manuscript of Marcos-Jubilar et al. with great interest and it provides a concise overview of (a part of) the relatively novel concept of immunothrombosis. However, I believe that for publication in the Journal of Clinical Medicine revisions are required.

Major points of concern:

-        The title of the manuscript does not cover the scope; the focus of the manuscript appears to be limited to the role of TF during immunothrombosis. Many other immunothrombosis processes (infectious or non-infectious) have been described that are not per se directly associated with TF, and these findings are not discussed here. I suggest to change the title to better capture the scope of the manuscript, and specify the scope of the review in the introduction, abstract, and conclusions.

-        A tendency to put TF on a pedestal continues throughout the manuscript, for instance on page 4 lines 102-104 (“Two key signals … through pyroptosis”): It may be true that the key signals that are described are indeed important, but in the current version it appears that only the TF-mediated processes are involved. This is not true and should be rewritten.

-        I would suggest the authors to separately discuss immunothrombosis as a result of different pathologies, possibly in a table. On page 2, lines 38-43 many pathologies are mentioned but are not further specified. Are there different TF-mediated immunothrombotic responses as a result of different pathologies?

-        Section 4 (Viral induced immunothrombosis): Please elaborate on the mechanism via which TF is induced and decrypted by viruses (lines 119-121) & on the statement why COVID-19 infection should be considered abnormal (lines 125-126). In a broader context, I do not fully grasp why viral-induced immunothrombosis is treated separately in a dedicated section, while in the manuscript it is stated that TF-mediated immunothrombosis is equally induced by bacterial infections or during another inflammatory status.

Minor remarks:

-        I believe that it would be helpful to ask a native English speaker to make some adjustments to the manuscript, which is well-written and clear but contains some grammatical errors.

-        Verify that all abbreviations are used correctly: Tissue factor (TF) is specified twice, TFPI and PRR are only used as abbreviation (in section 2).

-        Page 2 line 49: TF does not necessary interact with activated FVIIa, but also with FVII that significantly catalyzes its activation.

-        Could the authors elaborate on how platelets thrombin generation leads to platelet production of cytokines and NETosis?

-        Could the authors specify what TF decryption exactly entails?

-        Figure 1: Information is missing. How do PAMPs and PRRs mediate induction and decryption of TF? I believe that steps are missing. Also, in which (type of) cells do the processes take place, and is this process similar in every cell type?

-        Table 1: the table currently consists of several loose phrases and is not presented as an actual table. It is also missing the references to the specific studies, which can be added in an extra column.

-        References are lacking. For instance (not exclusively), page 1 lines 31-34: No reference for functioning of the intrinsic pathway. Page 3 lines 88-91: no reference for the “several studies”. Page 4 lines 111-116: For the HMGB1 studies just 1 reference is included.  

-        Include references to the original sources and not to review articles. For instance, on page 2 line 54 you make mention of TF release from different cell types. Throughout the manuscript, would it be possible to include references to the original studies in which certain processes were described?

Author Response

Thank you for your suggestions that undoubtedly will give more interest to this revision. I will answer to each question separately.

Major points of concern:

-        The title of the manuscript does not cover the scope; the focus of the manuscript appears to be limited to the role of TF during immunothrombosis. Many other immunothrombosis processes (infectious or non-infectious) have been described that are not per se directly associated with TF, and these findings are not discussed here. I suggest to change the title to better capture the scope of the manuscript, and specify the scope of the review in the introduction, abstract, and conclusions.

      With the new focus of the paper we think that we cover better more mechanisms related with immunothrombosis rather than tissue factor, so, we believe, that in this new version the title reflects the scope of the manuscript.

-        A tendency to put TF on a pedestal continues throughout the manuscript, for instance on page 4 lines 102-104 (“Two key signals … through pyroptosis”): It may be true that the key signals that are described are indeed important, but in the current version it appears that only the TF-mediated processes are involved. This is not true and should be rewritten.

      We tried in this new version not to be so focus to TF, covering better the role of different inflammation pathways. In order to avoid that feeling of putting TF like the main and only actor in inflammation process we have avoided such kind of expressions (key signal…)

-        I would suggest the authors to separately discuss immunothrombosis as a result of different pathologies, possibly in a table. On page 2, lines 38-43 many pathologies are mentioned but are not further specified. Are there different TF-mediated immunothrombotic responses as a result of different pathologies?

      As required we have developed this issue creating new separated sections of discussion: infectious disease (5) and other thromboinflammatory conditions (6) including arterial diseases (page 6) and autoimmune diseases (page 7).

-        Section 4 (Viral induced immunothrombosis): Please elaborate on the mechanism via which TF is induced and decrypted by viruses (lines 119-121) & on the statement why COVID-19 infection should be considered abnormal (lines 125-126). In a broader context, I do not fully grasp why viral-induced immunothrombosis is treated separately in a dedicated section, while in the manuscript it is stated that TF-mediated immunothrombosis is equally induced by bacterial infections or during another inflammatory status.

      We agree with the reviewer that creating a separated section focusing only in viral induced immunothrombosis but without addressing bacterial infections make the manuscript have some contradictions, so we have increased that section (page 5) to better explain that part.

Additionally, we have thoroughly explained covid-19 coagulopathy.

Minor remarks:

-        I believe that it would be helpful to ask a native English speaker to make some adjustments to the manuscript, which is well-written and clear but contains some grammatical errors.

      English has been reviewed.

-        Verify that all abbreviations are used correctly: Tissue factor (TF) is specified twice, TFPI and PRR are only used as abbreviation (in section 2).

Abbreviations has been reviewed through the document.

-        Page 2 line 49: TF does not necessary interact with activated FVIIa, but also with FVII that significantly catalyzes its activation.

It is explained in page 1 lines 29-37 how TF interacts with FVII making a complex at subendothelial level and how this interaction in pathological situations amplified the procoagulant TF activity around 100 times. Also, in pag 2 line 49 it has been added the interaction of TF not only with FVIIa but also FVII.

-        Could the authors elaborate on how platelets thrombin generation leads to platelet production of cytokines and NETosis?

A new section focusing on NETs role including the interplay of neutrophile and platelets have been included (page 4).

-        Could the authors specify what TF decryption exactly entails?

As suggested it is described in page 2, line 49-68.

-        Figure 1: Information is missing. How do PAMPs and PRRs mediate induction and decryption of TF? I believe that steps are missing. Also, in which (type of) cells do the processes take place, and is this process similar in every cell type?

Information to explain the table have been included.

Throughout the manuscript more details on the immunothrombosis process and induction of TF decryption have been included (specially in section 2 and 3), references to the image have been performed.

-        Table 1: the table currently consists of several loose phrases and is not presented as an actual table. It is also missing the references to the specific studies, which can be added in an extra column.

      As suggested table 1 has been improved and references to the studies have been added.

-        References are lacking. For instance (not exclusively), page 1 lines 31-34: No reference for functioning of the intrinsic pathway. Page 3 lines 88-91: no reference for the “several studies”. Page 4 lines 111-116: For the HMGB1 studies just 1 reference is included.  

-        Include references to the original sources and not to review articles. For instance, on page 2 line 54 you make mention of TF release from different cell types. Throughout the manuscript, would it be possible to include references to the original studies in which certain processes were described?

      In order to improve the article, as suggested, more references have been added and also original studies are being cited.

Round 2

Reviewer 3 Report

I assessed the revised manuscript of Marcos-Jubilar et al. with great interest and I would like to thank the authors for addressing all of the points raised. I have some minor remarks to improve readability of the manuscript.

Minor remarks:

-        For instance in line 35, the use of “as”: This should be changed to “such as”.

-        Line 58: It is not clear what the abbreviation PRR stands for

-        Line 65: “whether” should be “while”

-        Line 68/69: It is unclear what the authors with “the exposure of… cell surface”. Is exposed PS required for TF decryption? If so, PS exposed on what type of cell?

-        Line 70: “Unencrypted TF”, is this the same as decrypted TF? Please use the same term to avoid confusion

-        Line 75: “and” should be “via” or “by”

-        Line 87: I agree that TF is not the only important player during immunothrombosis, but I would rephrase this sentence that published data show that TF, together with e.g. monocytes or neutrophils, is a crucial player during immunothrombosis.

-        Line 124/125: PS externalization does not only favor TF decryption but also catalyzed the activation of coagulation proteins, which may also contribute to an increased thrombotic profile.

-        Line 143: “produced should be “induced”

-        Line 152: Histones are not present in DNA fragments but are one of the constituents of NETs (in addition to DNA).

-        Line 153: The abbreviation DAMPs has been introduced previously.

-        Section 5 (NETs and immunothrombosis): several references are missing to fundamental studies that have investigates NETs and the influence on thrombosis.

-        Line 165: “are recognized by”. These bacteria are recognized here by platelet TLR4? This should be added

-        Lines 167/168: How does LPS directly stimulate NET formation in the absence of platelets? Which receptors are involved?

-        Line 175: please rewrite this sentence.

-        Lines 184-187: The VWF string is not passively surrounded by platelets and neutrophils, but are actually initiating this process. This should be better described.

-        Line 205: Specify what is meant by “them”.

-        Line 222: The abbreviation interleukin has been used many times before in the manuscript.

-        Line 243/244: “In addition…inflammatory responses” comes out of the blue. Is this sentence related to IBD?

-        Lines 256-259: Many experimental studies have shown that hemostasis can be spared with effective thromboprotection, and the mentioned study by the authors is merely one example of this. This should be included in this passage.

-        Lines 262/263: It is strongly implied that in table 1 approaches are suggested that do not increase bleeding risk. This is not these case, since (regular) anticoagulation or antiplatelet therapy coincides with bleeding. The table should be differently introduced or the table should be modified.

-        Line 273: “Therapy focusing on targeting NETs is still far from reality”. Could the authors specify this statement? Do the authors believe that these drugs are not efficient enough, or have to many side effects?

-        Section 7D (targeting complement): During the course of the manuscript, the complement system has not been mentioned until line 304 where complement is introduced as a target to prevent immunothrombosis. I suggest to introduce the complement system earlier in the manuscript or not to discuss it here, to avoid confusion for the reader.

-        Line 330: In the conclusion, HMGB1 inhibition is mentioned as an attractive strategy to inhibit immunothrombosis. It is surprising that this strategy is not discussed in table 1. In addition, I believe that the last sentence of the conclusion should be a more “general” statement and not limited to one or two certain strategies.

Author Response

Thank you very much for the comments made on our work that will undoubtedly add scientific quality to the manuscript.

I will answer each of the suggestions made separately and I have highlighted all the changes made in the manuscript in red in order to find them easily.

Minor remarks:

-        For instance in line 35, the use of “as”: This should be changed to “such as”. OK

-        Line 58: It is not clear what the abbreviation PRR stands for. It has been modified for pathogen recognition receptors.

-        Line 65: “whether” should be “while”. OK

-        Line 68/69: It is unclear what the authors with “the exposure of… cell surface”. Is exposed PS required for TF decryption? If so, PS exposed on what type of cell?

We have added in order to clarify that it is exposed on endothelial cells.

-        Line 70: “Unencrypted TF”, is this the same as decrypted TF? Please use the same term to avoid confusion. Ok it is corrected to decrypted

-        Line 75: “and” should be “via” or “by”. OK

-        Line 87: I agree that TF is not the only important player during immunothrombosis, but I would rephrase this sentence that published data show that TF, together with e.g. monocytes or neutrophils, is a crucial player during immunothrombosis. We have change the sentences according to the suggestion.

-        Line 124/125: PS externalization does not only favor TF decryption but also catalyzed the activation of coagulation proteins, which may also contribute to an increased thrombotic profile. It has been modified as suggested.

-        Line 143: “produced should be “induced”. OK

-        Line 152: Histones are not present in DNA fragments but are one of the constituents of NETs (in addition to DNA). OK, this suggestion has been included in the text.

-        Line 153: The abbreviation DAMPs has been introduced previously. Long term has been cleared.

 -        Section 5 (NETs and immunothrombosis): several references are missing to fundamental studies that have investigates NETs and the influence on thrombosis.

Several references has been included to include some missing relevant paper focused in NET and thrombosis.

-        Line 165: “are recognized by”. These bacteria are recognized here by platelet TLR4? This should be added. OK

-        Lines 167/168: How does LPS directly stimulate NET formation in the absence of platelets? Which receptors are involved? As it is expressed in the following sentence COVID-19 activate NETs directly by neutrophil internalization and activating PAD4.

We have modified this sentence in order to develop how it occurs and which receptors are involved.

-        Line 175: please rewrite this sentence.

OK we have rewrite this sentence to clarify the meaning.

  •        Lines 184-187: The VWF string is not passively surrounded by platelets and neutrophils, but are actually initiating this process. This should be better described.

As previously, express, “pathogen and cytokines activate the endothelium producing a prothombotic state promoting multimeric chains of ultra-large von Willebrand factor (UL-VWF) release to endothelial cell surface where they will bind platelets”. But is is remarked that severe sepsis patients have ADAMTs13 levels reduced and it is associated with a reduction in the UL_VWF breaking that leads to a situation in which this long structures are surrounded, but of course, VWF has an active rol in Netosis.

-        Line 205: Specify what is meant by “them”

It has been added that it refers to cytokines

-        Line 222: The abbreviation interleukin has been used many times before in the manuscript. It has been eliminated.

-        Line 243/244: “In addition…inflammatory responses” comes out of the blue. Is this sentence related to IBD?

Yes it is related, as protein C pathway is important in the mucosal homeostasis and it is described that in IBD there is an impairment in this pathway associated to the spontaneous gut inflammation. We have added a sentence to clarify

-        Lines 256-259: Many experimental studies have shown that hemostasis can be spared with effective thromboprotection, and the mentioned study by the authors is merely one example of this. This should be included in this passage. As proposed, we have modified this part to include it.

-        Lines 262/263: It is strongly implied that in table 1 approaches are suggested that do not increase bleeding risk. This is not these case, since (regular) anticoagulation or antiplatelet therapy coincides with bleeding. The table should be differently introduced or the table should be modified.

We have include a new sentence to introduce the table: “we summarize in table 1 different therapeutic approaches for immunothrombosis:”

- Line 273: “Therapy focusing on targeting NETs is still far from reality”. Could the authors specify this statement? Do the authors believe that these drugs are not efficient enough, or have to many side effects?

We are going to modulate it as it is explained that DNase has been approved for nebulized treatment and it is used now in some trials endovenously after thrombectomy. However, those far from reality now, in our opinion, are PAD inhibitors, because although preclinical data are promising, there are very few in vivo data, however it is explained at lines 282-289.

- Section 7D (targeting complement): During the course of the manuscript, the complement system has not been mentioned until line 304 where complement is introduced as a target to prevent immunothrombosis. I suggest to introduce the complement system earlier in the manuscript or not to discuss it here, to avoid confusion for the reader

 It is not addressed thoroughly, however in section 6A (arterial diseases) it is mention: “NETs activate complement system leading to increased endothelial damage and upregulate platelet activation”

-   Line 330: In the conclusion, HMGB1 inhibition is mentioned as an attractive strategy to inhibit immunothrombosis. It is surprising that this strategy is not discussed in table 1. In addition, I believe that the last sentence of the conclusion should be a more “general” statement and not limited to one or two certain strategies

In order to include coherence and all the strategies address in the manuscript, HMBGB1 inhibition has been included in the table 1 and a new paragraph addressing their function is added. Last sentences have been modified. Conclusions have been modified to be less concrete.